# Garvicins AG1 and AG2: Two Novel Class IId Bacteriocins of *Lactococcus garvieae* Lg-Granada

**DOI:** 10.3390/ijms23094685

**Published:** 2022-04-23

**Authors:** Antonio Maldonado-Barragán, Estíbaliz Alegría-Carrasco, María del Mar Blanco, Ana Isabel Vela, José Francisco Fernández-Garayzábal, Juan Miguel Rodríguez, Alicia Gibello

**Affiliations:** 1Infection and Global Health Division, School of Medicine, University of St Andrews, St Andrews KY16 9TF, UK; 2Department of Animal Health, Veterinary School, Complutense University, Av. Puerta de Hierro, 28040 Madrid, Spain; ealegria@ucm.es (E.A.-C.); mmblanco@vet.ucm.es (M.d.M.B.); avela@vet.ucm.es (A.I.V.); garayzab@vet.ucm.es (J.F.F.-G.); gibelloa@vet.ucm.es (A.G.); 3VISAVET Health Surveillance Centre, Complutense University, Av. Puerta de Hierro, 28040 Madrid, Spain; 4Department of Nutrition and Food Science, Veterinary School, Complutense University, Av. Puerta de Hierro, 28040 Madrid, Spain; jmrodrig@vet.ucm.es

**Keywords:** antimicrobial peptides, in silico analysis, zoonotic diseases

## Abstract

*Lactococcus garvieae* causes infectious diseases in animals and is considered an emerging zoonotic pathogen involved in human clinical conditions. In silico analysis of plasmid pLG50 of *L. garvieae* Lg-Granada, an isolate from a patient with endocarditis, revealed the presence of two gene clusters (*orf*46–47 and *orf*48–49), each one encoding a novel putative bacteriocin, i.e., garvicin AG1 (GarAG1; *orf*46) and garvicin AG2 (GarAG2; *orf*48), and their corresponding immunity proteins (*orf*47 and *orf*49). The chemically synthesised bacteriocins GarAG1 and GarAG2 presented inhibitory activity against pathogenic *L. garvieae* strains, with AG2 also being active against *Listeria monocytogenes*, *Listeria ivanovii* and *Enterococcus faecalis*. Genetic organisation, amino acid sequences and antimicrobial activities of GarAG1 and GarAG2 indicate that they belong to linear non-pediocin-like one-peptide class IId bacteriocins. Gram-positive bacteria that were sensitive to GarAG2 were also able to ferment mannose, suggesting that this bacteriocin could use the mannose phosphotransferase transport system (Man-PTS) involved in mannose uptake as a receptor in sensitive strains. Intriguingly, GarAG1 and GarAG2 were highly active against their own host, *L. garvieae* Lg-Granada, which could be envisaged as a new strategy to combat pathogens via their own weapons.

## 1. Introduction

*Lactococcus garvieae* is a bacterial species that can be isolated from a wide spectrum of environmental, animal and food sources, including natural aquatic environments, sewage water, birds, fish, vegetables, meats, meat products and artisan dairy products [1,2,3,4]. In addition, this species is responsible for fish lactococcosis, a relevant infectious disease affecting different wild and farmed fish species [5,6], and has also been associated with some infectious conditions in domestic mammals [7,8,9]. In fact, *L. garvieae* is considered a potential zoonotic pathogen [10], as it is increasingly involved in a variety of human clinical conditions, from urinary tract infections and pneumonia to peritonitis, spondylodiscitis, acute cholecystitis, septicemia or endocarditis [11,12,13,14,15,16,17]. In recent years, the availability of more accurate methods for bacterial identification from clinical specimens, such as matrix-assisted laser desorption/ionization time-of-flight (MALDI-TOF) mass spectrometry (MS) and PCR amplification and sequencing of the 16S ribosomal DNA gene, has enabled the recognition of *L. garvieae* as an emerging agent of infective endocarditis involving both native and prosthetic valves [18,19,20,21,22,23,24].

Many bacterial strains are able to produce ribosomally synthesized antimicrobial peptides, so-called bacteriocins. The production of these antimicrobial peptides may contribute to bacterial fitness and may confer competitive advantages for survival and environmental spread [25,26]. In addition, bacteriocins have been proposed as an alternative to antibiotics due to concerns raised about antibiotic (multi)resistance among clinically relevant bacteria [25,27]. Seven garvicins (bacteriocins produced by *L. garvieae* strains) have been reported to date: garvicin L1-5 [28], garvicin ML [29], garvicin Q [30], garvicin A [31], garvicin KS [32], and garvicins B and C [33]. It was recently described that to exert their antimicrobial activity, garvicins A, B, C and Q (GarA, GarB, GarC and GarQ) can target a mannose-specific phosphotransferase system (Man-PTS) present on the membrane of sensitive strains [33,34].

*L. garvieae* Lg-Granada was originally isolated from the blood of a patient with endocarditis [20]. Although this strain does not exhibit bacteriocinogenic activity, sequencing of its plasmid pLG50 revealed the presence of two sets of genes (*orf46–47* and *orf48–49*), each one encoding a novel putative bacteriocin (*orf46* and *orf48*), and their respective immunity proteins (*orf47* and *orf49*) [35]. Because of the potential of garvicins as future antimicrobial agents in clinical and food settings, the objectives of this work were, first, to elucidate whether the bacteriocin-like peptides encoded by *orf46* and *orf48* actually displayed antimicrobial activity and, if so, to determine their respective antimicrobial spectra; and second, to assess the fermentation capacity of mannose of the sensitive indicator strains in order to determine whether the mannose phosphotransferase (Man-PTS) transporter system may be a possible cellular target for the recognition of these bacteriocins. 

## 2. Results

### 2.1. Analysis of Bacteriocin-Related Genes in the Sequence of Plasmid pLG50 of L. garvieae Lg-Granada

Both garvicin AG1 (GarAG1) and garvicin AG2 (GarAG2) belong to class IId (non-pediocin single linear peptides) bacteriocins [36]. GarAG1 is predicted to be synthesized as a 61-amino-acid precursor peptide that contains a 20-amino-acid double-glycine (GG) leader peptide, which, one processed, would render a mature bacteriocin peptide of 41 amino acids (Figure 1A). Although it shared high similarity with hypothetical proteins or putative bacteriocins of *Lactococcus petauri* and *Lactococcus garvieae* (Appendix A), the similarity with other known bacteriocins, such as garvicin Q (GarQ) and garvicin A (GarA) was low and mainly confined to the leader peptide. Thus, although the leader peptides of GarAG1, GarQ and GarA were 95% identical, no significant similarities were found between the mature peptides (Figure 1A).

GarAG2 is presumably synthesized as a 70-amino-acid precursor peptide with a double-glycine leader peptide of the GS type, where the second glycine is substituted by a serine residue; once processed, it would produce a mature bacteriocin peptide of 50 amino acids (Figure 1B). GarAG2 precursor peptide showed similarity with hypothetical proteins, putative pheromones/bacteriocins and known bacteriocins belonging to several species of the families *Enterococcaceae*, *Lactobacillaceae*, *Streptococcaceae* and *Leuconostocaceae* (Appendix A). Its closest known homologs (as precursor peptide bacteriocins) were garvicin Q, BacSJ, bovicin 255 and acidocin M, 84%, 46%, 42% and 46% shared identity, respectively. The GarAG2 leader peptide was similar to that of garvicin Q (75% shared identity) but showed low shared identity (<40%) with the leader peptides of BacSJ, acidocin M and bovicin 255 (Figure 1B). The amino acid sequence of mature GarAG2 showed 94%, 51%, 51% and 49% shared identity with the predicted mature peptides of garvicin Q, BacSJ, acidocin M and bovicin 255, respectively, sharing the N-terminal NGY and the central VTK conserved motifs (Figure 1B).

GarAG1 and GarAG2 shared 70% identity in their leader peptides, but their predicted mature peptides were quite different (<15% shared identity). Their alignment with known class IId bacteriocins that use the Man-PTS system as a receptor showed no appreciable conserved motifs (Figure 1C). Neither GarAG1 nor GarAG2 contained the pediocin-like motif YGNGVXC in their amino acid sequence.

Analysis of plasmid pLG50 revealed that the structural genes of precursor peptides GarAG1 (*orf46*, locus tag LA387_RS11080) and GarAG2 (*orf48*, locus tag LA387_RS11090) are located in close vicinity within the same locus (Figure 2). Just downstream of *orf46*, there is a gene (*orf47*, locus tag LA387_RS11085) that encodes a putative 99-aas protein (pI 9.70) showing 99% shared identity with hypothetical proteins of *Lactococcus petauri*. A putative promoter sequence (P1) containing the conserved −10 and −35 regions was found upstream of *orf46*. Downstream of *orf48*, there is a gene (*orf49*, locus tag LA387_RS11095) that encodes a putative 104-aas protein (pI 9.36) identical to the putative immunity protein of garvicin Q (AEN79391.1). A putative promoter sequence (P2) was found upstream of *orf48*, just overlapping the final coding sequence of *orf47*, whereas two inverted repeats of 8 bp (separated by 9 bp), which may function as a Rho-independent transcription terminator, were found just downstream of orf49. This genetic organization suggests that two putative transcripts could be generated: one driven by P1, which would include *orf46*, *orf47*, *orf48* and *orf49*; and one driven by P2, which would include *orf48* and *orf49* (Figure 2).

Upstream of *orf46*, there is a gene encoding a putative 185-aas protein that was identical to a recombinase family protein with a helix–turn–helix (HTH) domain of *Lactococcus garvieae* (WP_107106295). This protein is a serine recombinase similar to *Escherichia coli* transposon gamma–delta resolvase that catalyses the site-specific recombination of the transposon and also regulates its frequency of transposition (domain architecture ID 10133007). Downstream of *orf49*, there is a gene encoding a putative 228-aas protein that is virtually identical to IS6-like element IS1216 family transposase (protein family model: NBR001266) found in *Enterococcus*, *Lactococcus* and *Streptococcus*, among others (Figure 2).

### 2.2. Antimicrobial Activity of L. garvieae Lg-Granada

The cell-free supernatants (CFS) of *L. garvieae* Lg-Granada did not show antimicrobial activity against *L. garvieae* 8831, whereas those obtained from *L. garvieae* 21881, a garvicin-A-producing strain, led to inhibition halos >10 mm. Because no genes involved in precursor peptide processing and transport were found in plasmid pLG50, a fact that might impede its maturation and extracellular secretion, cell lysates of *L. garvieae* Lg-Granada were also tested. However, cell lysates showed no antimicrobial activity against the indicator strain, *L. garvieae* 8831. Therefore, it was decided to synthesize the peptides corresponding to the mature bacteriocins devoid of their potential leader sequences in order to determine whether they actually possess antimicrobial activity.

### 2.3. Antimicrobial Activity of the Chemically Synthesized Garvicins AG1 and AG2

Both peptides, GarAG1 and GarAG2, showed antimicrobial activity against some of the Gram-positive indicator strains tested in this work (Table 1). However, the antimicrobial spectrum of the two garvicins was very different. The spectrum of GarAG1 was narrow and active against *L. garvieae* (16 strains inhibited of 18 tested; 16/18) and *Pediococcus acidilactici* (1/1), whereas GarAG2 showed a wider spectrum of activity and included strains belonging to different genera and species, such as *Carnobacterium maltaromaticum* (1/1), *Enterococcus faecalis* (3/3), *L. garvieae* (16/18), *Lactococcus lactis* (2/2), *Listeria ivanovii* (1/1), *Listeria monocytogenes* (1/1) and *Streptococcus salivarius* (1/1).

In addition, GarAG2 was more active than GarAG1 for those indicator strains that were susceptible to both garvicins, with the only exception of *P. acidilactici* CECT 98, which was sensitive only to GarAG1. Hence, the inhibition zones displayed by GarAG2 and GarAG1 against the susceptible *L. garvieae* strains ranged from 12 to 22 mm and 9 to 16 mm diameter, respectively. None of the garvicins showed activity against the two Gram-negative strains (*E. coli* CECT 515T and *Salmonella enterica* S79) included in the assays.

It must be highlighted that despite the fact that the plasmid pLG50 of *L. garvieae* Lg-Granada contains immunity genes for GarAG1 (*orf47*) and GarAG2 (*orf49*), this strain was sensitive to both garvicins. In contrast, the garvicin-A-producing *L. garvieae* 21881 strain and its derivative, *L. garvieae* 21881-N, which lacks all the genes required for garvicin A production and immunity, were resistant to both GarAG1 and GarAG2.

Garvicins AG1 and AG2 did not display additive or synergistic antimicrobial activities, as the size (18 mm) of the inhibition halo of the disk containing both GarAG1 (2.5 μg) and GarAG2 (2.5 μg) was identical to that displayed by GarAG2 alone (2.5 μg) against *L. garvieae* 8831. In addition, the sizes of the inhibition zones when two disks containing the same concentrations (2.5 μg) of GarAG1 (14 mm) and GarAG2 (18 mm) were placed immediately adjacent on each other were identical to those displayed by the respective garvicins when each disk was placed in a different location within the same plate. The minimal inhibitory concentration (MIC) value of GarAG1 was 5 μg/mL against *L. garvieae* 8831, whereas those of GarAG2 were 2.5, 0.1 and 0.1 μg/mL against *L. garvieae* 8831, *E. faecalis* OEA1 and *L. monocytogenes* 51112, respectively. These data are in agreement with the inhibition halos displayed by GarAG1 and GarAG2 against such indicator strains.

### 2.4. Relationship between Sensitivity to Garvicins AG1 and AG2 and Mannose Fermentation Ability

No significant relationship was found between the sensitivity of the indicator strains to GarG1 and their ability to ferment mannose (Table 1). Conversely, all the Gram-positive indicator strains that were sensitive to GarAG2 (25 out of 34 strains) were able to ferment mannose. In addition, most of the Gram-positive indicator strains that fermented mannose (28 out of 34 strains) were sensitive to GarAG2 (Table 1). The only exceptions were *S. aureus* CECT 86^T^, *L. garvieae* 21881 (garvicin A producer) and its derivative, *L. garvieae* 21881-N (garvicin A non-producer), which showed a mannose fermentation ability but were resistant to GarAG2. In contrast, the rest of the Gram-positive strains that were resistant to GarAG2 (*Aerococcus viridans* CECT 978^T^, *Bacillus cereus* CECT 5050^T^, *P. acidilactici* CECT 98, *Streptococcus agalactiae* MP007, *Streptococcus parauberis* CCUG 39954^T^ and *Streptococcus uberis* CECT 994^T^) did not ferment mannose (Table 1).

## 3. Discussion

In this work, in silico analysis of the gene sequence of plasmid pLG50 of *L. garvieae* Lg-Granada allowed for the identification of the structural and immunity genes of two novel bacteriocins: garvicins AG1 and AG2. Based on the genetic organisation of the genes encoding GarAG1 and GarAG2, their corresponding deduced amino acid sequences, which lack the conserved pediocin-like motif YGNGVXC, and their non-synergistic antimicrobial activity, both garvicins are considered to belong to the class IId one-peptide bacteriocins [36]. Although two promoters (P1 and P2) were identified and, in theory, two putative transcripts could be generated, one driven by P1, (*orf46*-*orf47*-*orf48*-*orf49*) and the other by P2 (*orf48*-*orf49*) (Figure 2), bacteriocinogenic activity was not detected either in the CFS or in the cell lysates obtained from *L. garvieae* Lg-Granada. The fact that *L. garvieae* Lg-Granada was very sensitive to both bacteriocins could indicate a lack of expression of these genes (including the immunity genes). Lactic acid bacteria frequently contain silent (cryptic) bacteriocins that cannot be produced by the host cell because of a lack of key genes required for their biosynthesis, regulation, processing or transport or even because of the absence of the appropriate social environment [26,37].

In relation to garvicins, it has been previously found that *L. garvieae* 21881 not only produces garvicin A [31] but also contains the structural genes of two unexpressed bacteriocins, garvicin B (GarB) and garvicin C (GarC) (accession no. WP_014386584.1 and WP_014386275.1, respectively), which are encoded in its plasmids, pGL2 and pGL1, respectively [33]. The authors hypothesized that they are not expressed because of a lack of genes required for their secretion. However (and similarly to GarAG1 and AG2), the biochemically synthesized GarB and GarC bacteriocins showed antimicrobial activity.

The existence in plasmid pLG50 of a transposase-encoding insertion sequence (IS1216) downstream of *orf48*-*orf49*, together with the transposon resolvase located upstream of *orf46*-*orf47*, resembles the structure of the Tn1546 transposon that carries vancomycin-resistant genes [38]. This could indicate that the garvicin AG1 and AG2 gene clusters could have been recently acquired by *L. garvieae* Lg-Granada through horizontal transfer of plasmid pLG50 or through independent transposition events related to IS1216, lacking the genes needed for its production, processing or secretion. In this regard, the absence of a dedicated ATP-binding cassette (ABC) transporter could prevent the processing of the double-glycine (GG) leader peptide and the secretion of the mature (active) garvicins, thus explaining the lack of antimicrobial activity displayed by *L. garvieae* Lg-Granada. The lack of a dedicated ABC transporter could also explain the sensitivity of *L. garvieae* Lg-Granada to GarAG1 and GarAG2, as these transporters can be involved in providing immunity to bacteriocins [39].

Similarly to GarA, GarB and Gar C [31,33], GarAG1 has a narrow antimicrobial spectrum, mostly limited to *L. garvieae* strains. In contrast, the antimicrobial spectrum of GarAG2 is wider, and in addition to species of the genus *Lactococcus*, it targets strains of the genera *Enterococcus*, *Streptococcus* and *Listeria*. The spectrum of GarAG2 seems very similar to that of GarQ, including the lack of sensitivity of *Pediococcus* strains [30,34], which could be explained by the high amino acid identity (94%) between the two garvicins.

The Man-PTS transport system [40] is used by subclass IIa bacteriocins (pediocin PA-1-like bacteriocins) and subclass IId bacteriocins, including GarA, GarB, GarC and GarQ [33,34], as a receptor in sensitive bacterial cells. Among other functions, this system is responsible for mannose uptake and phosphorylation. Interestingly, all the Gram-positive indicator strains that were sensitive to GarAG2 were also able to ferment mannose, whereas those that did not ferment this sugar were resistant to the bacteriocin. These results, together with the high amino acid sequence similarity between GarAG2 and GarQ, suggest that GarAG2 most probably also uses the Man-PTS system as its main target in bacterial cells. This is an interesting property as a tool for bacterial control because eukaryotic cells lack such a system [41]. On the contrary, we did not find a relationship between GarAG1 activity and the ability of the sensitive strains to ferment mannose, indicating that GarAG1 does not target the Man-PTS system to exert its antimicrobial activity.

Among the Gram-positive indicator strains, *S. aureus* CECT 86^T^, *L. garvieae* 21881 and *L. garvieae* 21881-N (mannose-positive but resistant to GarAG2) were the only exceptions to this association between mannose fermentation and sensitivity to GarAG2. Cross-immunity between GarA and GarAG2 might be an explanation for this phenotype, as *L. garvieae* 21881 produces GarA and expresses its corresponding immunity gene. However, this is not a valid explanation because *L. garvieae* 21881-N is a derivative of *L. garvieae* 21881 that lacks the pGL5 plasmid containing the genes involved in garvicin A biosynthesis, immunity, processing and transport [31]. A plausible explanation could be the fact that both *L. garvieae* 21881 and its derivative, *L. garvieae* 21881-N, harbour the plasmids pGL1 and pGL2, which encode garvicins GarC and GarB (and their corresponding immunity genes), respectively, thus providing cross-immunity to both GarAG1 and AG2. Although the mechanism of resistance to GarAG2 of *L. garvieae* 21881 is currently unknown, resistance to the action of a class IIa or class IId bacteriocin linked to the preservation of the ability to ferment mannose has already been described [34].

The two Gram-negative strains tested in this study were able to ferment mannose but were resistant to GarAG2. It must be highlighted that no lactococcal bacteriocin has antimicrobial activity against intact Gram-negative bacteria, as the outer membranes of these bacteria prevents the interaction between the bacteriocin and its receptors, which are located in the plasmatic membrane [42].

Both GarAG1 and GarAG2 have remarkable activity against *L. garvieae* strains, which renders these bacteriocins attractive tools to control this species in aquaculture systems. In addition, the spectrum of GarAG2 includes *L. monocytogenes*, a very relevant foodborne pathogen and, as a consequence, one of the main targets in bacteriocin research [43]. However, we are aware of the many limitations that these bacteriocins face for practical applications. Because *L. garvieae* Lg-Granada harbours the structural gene for GarAG1 and GarAG2 but does not express them, their heterologous expression at high levels in safe strains is one approach for the application of these bacteriocins in clinical settings or food systems [44]. However, the lack of genes encoding their processing and secretion would make necessary the use of the secretory machinery of other class IId bacteriocins, such as that of GarA or GarQ [33]. In this context, the use of a natural GarA producer strain (in the case of GarAG1) or a natural GarQ producer strain (in the case of GarAG2) may be a more practical approach, given the similarities between their respective antimicrobial spectra. Finally, another way to use the potential of these bacteriocins is through the use of synthetic peptides in a pharma-like setting, although this would require a considerable investment and regulatory changes. In conclusion, this study confirms that mining the increasing number of bacterial genomes is an excellent strategy for the discovery of active but not expressed novel bacteriocins and may introduce several surprises in the future. To our knowledge, this is the first study reporting the discovery of bacteriocins that are active against the pathogenic bacteria that harbour them, which can be envisaged as a strategy to combat pathogens with their own weapons.

## 4. Materials and Methods

### 4.1. Bacterial Strains and Growth Conditions

Bacteria used in this study included pathogenic isolates of veterinary and clinical relevance isolated from food, animals and humans, as well as other non-pathogenic isolates involved in food fermentations (Table 1). *Lactococcus garvieae* Lg-Granada was originally isolated from a patient with endocarditis in the mitral valve [20]. *Enterococcus*, *Listeria*, *Streptococcus* and *Staphylococcus* strains were routinely grown in brain heart infusion (BHI; Oxoid, Basingstoke, UK); *Carnobacterium*, *Lactococcus* and *Pediococcus* strains were grown in de Man, Rogosa and Sharpe (MRS) medium (Oxoid, Basingstoke, UK); and *Escherichia*, and *Salmonella* strains were grown in Luria-Bertani (LB) medium (Table 1). The samples were maintained as frozen stocks at −80 °C in MRS or BHI plus 20% (*vol/vol*) glycerol. All strains were incubated for 24 h at 30 °C without agitation, except for Gram-negative strains, which were grown at 37 °C with agitation.

### 4.2. DNA and Amino Acid Sequence Analysis

Searches for potential DNA and amino acid similarities of plasmid pLG50 of *L. garvieae* Lg-Granada (GenBank acc. no. CP084378.1; RefSeq acc. no. NZ_CP084378.1) in nucleotide and protein databases were carried out using the Basic Local Alignment Search Tool (BLAST; http://blastncbi.nlm.nih.gov/; access date: 1 December 2021) [45]. Promoter sequences were predicted with Softberry BProm (Softberry, Inc., Mount Kisco, NY, USA; www.softberry.com; access date: 1 December 2021) [46]. Rho-independent terminators were detected with the ARNold Web server (http://rssf.i2bc.paris-saclay.fr/toolbox/arnold/; access date: 1 December 2021) [47]. The MultAlin multiple sequence alignment interface page (http://multalin.toulouse.inra.fr/multalin/; access date: 1 December 2021) was used for alignment of the amino acid sequences [48]. Finally, the WinPep program (Lars Hennig, Freiburg, Germany) [49] was used for physicochemical analysis of peptides (isoelectric point, molecular weight).

### 4.3. Preparation of Garvicins AG1 and AG2

The putative mature bacteriocins encoded by *orf46* (GRETLAQDIKRVYDSIWPNDTAWYTGKNNKTNIPPYSPYGH) and *orf48* (GTPLFYGANGYLTRENGKYVYRVTKDPVSAVFGVISNGWGSAGAGFGPQH), termed garvicin AG1 (GarAG1) and AG2 (GarAG2), respectively, after Prof. Alicia Gibello, were chemically synthesized by GenScript Biotech (Piscataway, NJ, USA), with >95% purity. None of the synthesized peptides was formylated. The peptides were solubilized to concentrations of 1 and 0.1 mg/mL in 0.1% (*vol/vol*) trifluoroacetic acid (Sigma-Aldrich, Madrid, Spain) and stored at −20 °C until use.

### 4.4. Bacteriocin Assays

Initially, the antimicrobial activity of the cell-free supernatants (CFS) and cell lysates of *L. garvieae* Lg-Granada was assessed by an agar diffusion test [31] using *L. garvieae* 8831 as the indicator strain and *L. garvieae* 21881 (a garvicin-A-producing strain) and its derivative, *L. garvieae* 21881-N (garvicin A non-producer), as positive and negative control, respectively. 

CFS were prepared by centrifuging 2 mL of 16 h overnight cultures of *L. garvieae* Lg-Granada at 10,000× *g* for 3 min. Then, the supernatants were filtered through cellulose acetate membrane filters with a 0.22 μm pore size (Sartorius, Göttingen, Germany). To prepare the cell lysates, the cell pellets from the overnight cultures were resuspended in 100 μL of sterile MRS and separated into two fractions. Sterile glass beads were added to the first fraction, and a lysis matrix (MP Biomedicals, Solon, OH, USA) was added to the second fraction. Subsequently, both fractions were lysed by 3 cycles of FastPrep (MP Biomedicals, Solon, OH, USA) at a speed of 4 m/s for 20 s. Wells (diameter, 4.0 mm) were created with a sterile cork borer on the agar plates where the indicator strains had been inoculated. Each well was filled with either 100 μL of a CFS or 100 μL of a cell lysate. Plates were incubated at 30 °C for 16 h, and then the diameters of the inhibition zones were recorded.

The antimicrobial activity of garvicins AG1 and AG2 was assayed by an agar diffusion test in which wells were replaced by sterile disks impregnated with 2.5 μg of the tested peptide. All the species and strains used as indicator microorganisms to determine the spectrum of activity of the bacteriocins are listed in Table 1. *L. garvieae* 8831 was used as the indicator strain to assess whether either of the garvicins might have synergistic or additive activity. In this case, disks containing each bacteriocin (GarAG1 or GarAG2) were placed adjacent to each other on the same plate, and disks containing both bacteriocins (GarAG1 plus GarAG2) were also included in the assays; their inhibition halos were compared with those observed for non-adjacent disks containing each garvicin. 

To determine the minimal inhibitory concentration (MIC) of garvicins AG1 and AG2, three sensitive indicator strains (*L. garvieae* 8831, *Listeria monocytogenes* 51112 and *Enterococcus faecalis* OEA1) were used (Table 1). For this purpose, the indicator stains were grown in Mueller–Hinton broth (MH; Fisher Scientific, Pittsburgh, PA, USA) and inoculated in 96-well microtiter plates at a concentration of 1.5 × 10^6^ colony forming units per millilitre (CFU/mL). Each bacteriocin was tested at the following final concentrations: 5, 4, 3, 2.5, 2, 1.5, 1, 0.5, 0.25, 0.1 and 0 μg/mL.

### 4.5. Mannose Fermentation Ability among the Indicator Strains

The amino acid sequence of mature GarAG2 showed a very high degree of identity with garvicin Q (see below), a bacteriocin that uses the Man-PTS system to recognize target cells [34]. As a consequence and in order to elucidate whether the Man-PTS system may be a possible cellular target for the recognition of garvicin AG2, mannose fermentation ability of the indicator strains was tested in mannose-phenol red broth (tryptone, 10 g/L; mannose, 10 g/L; sodium chloride, 5 g/L; phenol red, 0.018 g/L). Phenol red acts as a pH indicator, staying red in the 6.8–8 range and turning yellow at acidic pH, indicating fermentation. The tubes were incubated at 37 °C for 24 h, and a non-inoculated tube was included as a negative fermentation control.

## Figures and Tables

**Figure 1 ijms-23-04685-f001:**
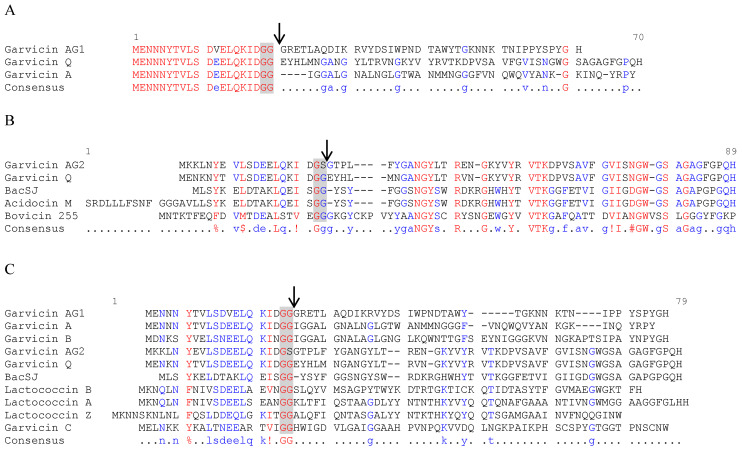
Alignment of the amino acid sequences of garvicin AG1 (**A**) and garvicin AG2 (**B**) precursor peptides with their known homologs and with class IId bacteriocins that use a mannose-specific phosphotransferase system (Man-PTS) as a receptor (**C**). The double-glycine motifs (GG or GS) of the leader peptides are highlighted with a grey background, and the black arrow indicates the cleavage site that generates the mature peptide bacteriocins. The sequences were aligned using the Multalin interface page (http://multalin.toulouse.inra.fr/multalin/ accessed on 1 December 2021). Highly conserved residues (consensus level = 90%) are indicated in red, whereas weakly conserved residues (consensus level = 50%) are indicated in blue. Consensus symbols are: !, I or V; $, L or M; %, F or Y; #, N, D, Q, E, B or Z. NCBI reference sequences, UniProtKB/Swiss-Prot or GenBank accession numbers: Garvicin AG1, WP_225667055.1 (*Lactococcus garvieae* Lg-Granada, plasmid); Garvicin AG2, WP_165719065.1 (*Lactococcus garvieae* Lg-Granada, plasmid); Garvicin A, CCF71073.1 (*Lactococcus garvieae* 21881, plasmid); Garvicin Q, AEN79392.1 (*Lactococcus garvieae* BCC 43578, plasmid); Bacteriocin BacSJ, CAR92206.2 (*Lacticaseibacillus paracasei* subsp. *paracasei* BGSJ2-8, plasmid); Acidocin M, partial, BAB86318.1 (*Lactobacillus acidophilus* TK8912, plasmid); Bovicin 255, AAG29818.1 (*Streptococcus* sp. LRC 0255, chromosome); Garvicin B, CCF55365.1 (*Lactococcus garvieae* 21881, plasmid); Garvicin C, CCF55362.1 (*Lactococcus garvieae* 21881, plasmid); Lactococcin A, P0A313.1 (*Lactococcus lactis* subsp. *cremoris* 9B4, plasmid); Lactococcin B, P35518.1 (*Lactococcus lactis* subsp. *cremoris* 9B4, plasmid); Lactococcin Z, BAU29928.1 (*Lactococcus lactis* QU7, chromosome).

**Figure 2 ijms-23-04685-f002:**
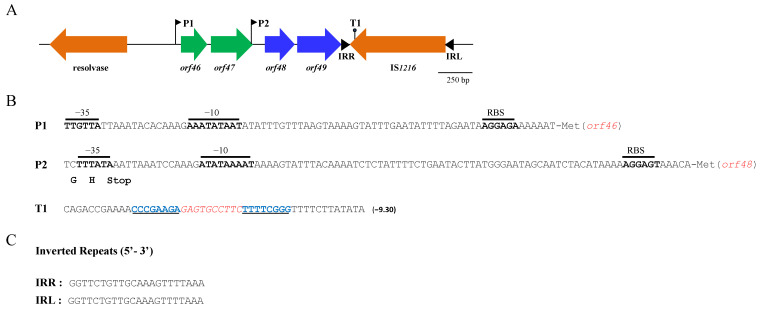
Schematic representation of a 3.1-kpb DNA fragment from plasmid pLG50 of *L. garvieae* Lg-Granada that contains the genes encoding garvicins AG1 (*orf46*) and AG2 (*orf48*) and their corresponding immunity proteins, *orf47* and *orf49*, respectively (**A**); detailed analysis of predicted promoters and Rho-independent terminator (**B**); and inverted repeats sequences of the IS1216 transposase (**C**). (**A**) P1 and P2 are putative promoter sequences, whereas T1 is a putative Rho-independent transcription terminator. IRR and IRL correspond to the right and left inverted repeat sequences flanking the transposase of IS1216, respectively. Resolvase indicates the *orf* encoding a putative transposon gamma–delta resolvase gene. (**B**) The putative promoters P1 and P2 were detected with Softberry BProm. The typical −35 and −10 boxes and the ribosome binding sites (RBS) are shown; “Met” in the P1 and P2 sequences indicates the methionine residues of GarAG1 and AG2, respectively. “Stop” in the P2 sequence indicates the termination codon of *orf46*. ARNold was used to predict the Rho-independent terminator (T1); base pairs of the hairpin are shown in blue boldface, and apical loops are indicated by red italics. The predicted free energy of terminator hairpins (kcal/mol) is given in parentheses. (**C**) The 22 bp right and left inverted repeats (IRR and IRL, respectively) of IS1216 showed 100% identity in their nucleotide sequence.

**Table 1 ijms-23-04685-t001:** Antimicrobial spectrum of garvicins AG1 and AG2, as well as the mannose-fermenting ability of the indicator strains.

Indicator Strain	Origin	Source ^1^	Inhibition (mm) ^2^ by:		Man ^3^
			Garvicin AG1	Garvicin AG2	
*Aerococcus viridans* CECT 978^T 4^	Air sample	CECT	−	−	−
*Bacillus cereus* CECT 5050^T^	Unknown	CECT	−	−	−
*Carnobacterium maltaromaticum* 02/5685	Fish (trout)	FVM-S	−	18	+
*Enterococcus faecalis* MP48	Human (vagina)	FVM-N	−	13	+
*E. faecalis* EIPO	Human (urine)	FVM-N	−	18	+
*E. faecalis* OEA1	Human (urine)	FVM-N	−	20	+
*Escherichia coli* CECT 515^T^	Human (urine)	CECT	−	−	+
*Lactococcus garvieae* 19	Human (urine)	HPM	12	16	+
*L. garvieae* 21881 (GarA+) ^5^	Human (blood)	HRV	−	−	+
*L. garvieae* 21881-N (GarA−) ^6^	Human (blood)	FVM-S	−	−	+
*L. garvieae* 3AA7	Food (cheese)	IPLA	12	14	+
*L. garvieae* 57	Fish (trout)	VISAVET	12	18	+
*L. garvieae* 65	Fish (trout)	VISAVET	16	16	+
*L. garvieae* 80	Fish (trout)	VISAVET	14	17	+
*L. garvieae* 85	Fish (trout)	VISAVET	10	12	+
*L. garvieae* 8831	Fish (trout)	VISAVET	14	18	+
*L. garvieae* BM06/00349	Human (blood)	HRV	12	22	+
*L. garvieae* CAS-2	Food (cheese)	IPLA	14	15	+
*L. garvieae* CECT 4531^T^	Bovine	CECT	14	18	+
*L. garvieae* CP-1	Fish (trout)	FVM-S	10	14	+
*L. garvieae* Lg-Granada	Human (blood)	HVN	13	16	+
*L. garvieae* Lg-Granada 240-88	Human (blood)	HVN	12	20	+
*L. garvieae* N-201	Food (cheese)	IPLA	10	18	+
*L. garvieae* NRTC 0607	Food (vegetable)	HUJ	9	20	+
*L. garvieae* T2-17	Food	IPLA	12	18	+
*Lactococcus lactis* subsp. *lactis* MG1363	Food	IFR(QI)	−	19	+
*L. lactis* subsp. *lactis* MP29	Food (cheese)	FVM-N	−	29	+
*Listeria ivanovii* CECT 913^T^	Ovine	CECT	−	24	+
*Listeria monocytogenes* 51112	Food	FVM-S	−	25	+
*Pediococcus acidilactici* CECT 98	Food	CECT	14	−	−
*Salmonella enterica* S79	Poultry (faeces)	FVM-S	−	−	+
*Staphylococcus aureus* CECT 86^T^	Human (pleural fluid)	CECT	−	−	+
*Streptococcus agalactiae* MP007	Human (vagina)	FVM-N	−	−	−
*Streptococcus parauberis* CCUG 39954^T^	Bovine (mastitis)	CCUG	−	−	−
*Streptococcus salivarius* CECT 805^T^	Human (blood)	CECT	−	10	+
*Streptococcus uberis* CECT 994^T^	Bovine (mastitis)	CECT	−	−	−

**^1^** CECT, Colección Española de Cultivos Tipo (Universidad de Valencia, Burjasot, Spain); CCUG, culture collection of the University of Gothenburg (University of Göteborg, Sweden); FVM-N, Department of Nutrition and Food Science, Facultad de Veterinaria, Universidad Complutense (Madrid, Spain); FVM-S, Departamento de Sanidad Animal, Facultad de Veterinaria, Universidad Complutense (Madrid, Spain); HPM, Hospital Puerta del Mar (Cádiz, Spain); HVN, Hospital Virgen de las Nieves (Granada, Spain); IPLA, Instituto de Productos Lácteos de Asturias (Villaviciosa, Spain); HUJ, Hiroshima University Japan (Japan); HRV, Hospital Royo Villanova (Zaragoza, Spain); IFR (QI), Institute of Food Research (Quadram Institute) (Norwich, United Kingdom); VISAVET, Centro de Vigilancia Sanitaria (Madrid, Spain). ^2^ Diameter of the halo of inhibition (mm). −, no inhibition. Antimicrobial activity was determined by the agar diffusion test using sterile disks impregnated with 2.5 μg of the tested bacteriocin. ^3^ Ability to ferment mannose. −, no fermentation; +, fermentation. ^4^ “^T^”, type strain. ^5^ Produces garvicin A (GarA+). ^6^ Derivative of *L. garvieae* 21881 lacking the plasmid pGL5 that encodes garvicin A; does not produce garvicin A (GarA−).

## Data Availability

The genome sequence of plasmid pLG50 of *Lactococcus garvieae* Lg-Granada (35) is deposited in GenBank under the accession numbers CP084378.1 and NZ_CP084378.1.

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
