# Peer review of "Garvicins AG1 and AG2: Two Novel Class IId Bacteriocins of Lactococcus garvieae Lg-Granada"

_ijms, 2022, doi:10.3390/ijms23094685_

Round 1
Reviewer 1 Report
This MS. Described the identification of two novel class IId bacteriocins from Lactococcus garvieae. The topic and the proposed conclusion of this paper are interesting. However, there are some major concerns needs to be addressed. The authors define AG1 and AG2 as silent bacteriocin simply because no activity was detected from CFS. Could it be due to low expression level? Did you try to concentrate the CFS and then test the activity.
The speculated reason is the lack of genes encoding processing enzymes in the cluster. Since both structural genes have strong promoter as shown in Figure 2. It would be necessary to analysis the expression of these genes at least at transcription level.
Reviewer 2 Report
Dear authors,
This is an interesting study on Lactococcus garvieae bacteriocins and their antimicrobial activity.
Some of my comments are listed below:
All keywords have been already listed in the title of the manuscript. Could the author consider replacing those to characterize the main topic of research or methodology-specific term?
Please italicize in silico (line 18,... ) L. garvieae (line 147,226, 264)
Line 44-46. Could the authors replace the "tools" with "methods" or methodology? Please provide the full names for the abbreviation at the first mention. Culturomics includes the methods were listed to illustrate the accuracy of microbial identification.
2.3. subsection. Please describe in brief the range of antimicrobial activity for the tested garvicin AG1 and AG2 with an indication of specific microorganisms and the inhibition zones.
Line 184-187. Could the authors rewrite the sentence without parenthesis, not clear enough. Please avoid extensively use of parenthesis throughout the manuscript (e.g. 166-167, 170, 234, etc)
Line 202-205. Please rewrite the sentence, not clear enough.
Line 206. “its derivates”. What do the authors think with “derivates”?
Line 269-272. Please merges the sentences.
Please add the appropriate reference on the impact of the outer membrane of gram-negative bacteria.
4.1. Bacterial strains and growth conditions. Could the authors describe in one sentence the origin of isolates? Were they the isolates of clinical and/ or veterinary importance, environmental isolates? Please provide the details on how the isolates were maintained prior to the experiment? Add the duration of culturing for describing of growth conditions.
4.4. Bacteriocin assay. Line 359. What was a negative control?
In how many replicates were the antimicrobial essays performed?
Round 2
Reviewer 1 Report
As a short communication. The current version can be be acceptted.
Author Response
We would like to thank Reviewer 1 for his/her suggestions and constructive criticism for improving the manuscript.
This work was initially sent as "Article" but after the consideration of the Editor we agreed to resubmit it as short Communication, which also agrees with the suggestion of Reviewer 1.